# The Role of the Bone Morphogenetic Protein Antagonist Noggin in Nucleus Pulposus Intervertebral Disc Cells

**DOI:** 10.3390/ijms252111803

**Published:** 2024-11-02

**Authors:** Shuimu Chen, Sebastian Bigdon, Carsten Riether, Xiaochi Ma, Xiaoyi Niu, Sonja Häckel, Zhen Li, Benjamin Gantenbein

**Affiliations:** 1Tissue Engineering for Orthopedics & Mechanobiology (TOM), Bone & Joint Program, Department for BioMedical Research (DBMR), Faculty of Medicine, University of Bern, CH-3008 Bern, Switzerland; shuimu.chen@unibe.ch; 2Graduate School for Cellular and Biomedical Sciences (GCB), University of Bern, CH-3012 Bern, Switzerland; 3Department of Orthopaedic Surgery and Traumatology, Inselspital, Bern University Hospital, University of Bern, CH-3010 Bern, Switzerland; sebastian.bigdon@insel.ch (S.B.); sonja.haeckel@insel.ch (S.H.); 4Department of Medical Oncology, Inselspital, Bern University Hospital, University of Bern, CH-3010 Bern, Switzerland; carsten.riether@unibe.ch; 5Department for BioMedical Research, University of Bern, CH-3008 Bern, Switzerland; xiaochi.ma@unibe.ch (X.M.); xiaoyi.niu@outlook.com (X.N.); 6Graduate School for Health Sciences (GSH), University of Bern, CH-3012 Bern, Switzerland; 7AO Research Institute Davos, CH-7270 Davos, Switzerland; zhen.li@aofoundation.org

**Keywords:** low back pain, human intervertebral disc, degeneration, nucleus pulposus cells, Noggin

## Abstract

Low back pain (LBP) is a significant global health issue, contributing to disability and socioeconomic burdens worldwide. The degeneration of the human intervertebral disc (IVD) is a critical factor in the pathogenesis of LBP. Recent studies have emphasized the significance of a specific set of genes and extracellular matrix (ECM) in IVD health. In particular, Noggin has emerged as a critical gene due to its high expression levels in healthy nucleus pulposus cells (NPCs) observed in our previous research. In this study, it was hypothesized that decreased Noggin expression in NPCs is associated with IVD degeneration and contributes to LBP development. A lentivirus-mediated RNAi was applied to knock down Noggin expression in primary NPCs from six human donors. The NPCs after transduction were evaluated through cell viability analysis, XTT assay, and cell apoptosis analyses. After two weeks, a colony formation assay was used to examine the anchor-independent growth ability of transduced cells. At the transcript level, anabolic and catabolic markers were quantified using RT-qPCR. The results demonstrated that lentivirus-mediated downregulation of Noggin significantly inhibited cell proliferation, reduced cell viability, and suppressed colony formation, while inducing apoptosis in human NPCs in vitro. Notably, it disrupted cellular anabolic processes and promoted catabolic activity in human NPCs post-transduction. Our findings indicated that the degeneration of human IVD is possibly related to decreased Noggin expression in NPCs. This research provides valuable insights into the role of Noggin in IVD homeostasis and its implications in LBP treatment.

## 1. Introduction

Low back pain (LBP) represents a significant global health issue [1], contributing to substantial disability and socioeconomic impact due to its prevalence and the associated healthcare costs [2,3]. The degeneration of the human intervertebral disc (IVD) is central to the pathogenesis of LBP [4,5]. Understanding the molecular mechanisms underlying IVD degeneration is essential for developing effective therapeutic strategies to alleviate LBP and enhance patient quality of life.

Intervertebral discs comprise a gelatinous nucleus pulposus (NP) surrounded by a stress-resistant annulus fibrosus (AF) [6,7,8]. The NP is crucial for the disc’s ability to absorb shocks and distribute mechanical loads [9]. Nucleus pulposus cells (NPCs), derived from the NP, play a vital role in maintaining the health and functionality of the IVD by producing extracellular matrix (ECM) components [10,11]. Recent studies [12,13,14,15] have highlighted the significance of various genes and proteins in maintaining IVD health. Among these, Noggin has emerged as a critical factor due to its high expression levels in healthy NPCs, as observed in previous research [16].

Noggin is a well-known antagonist of bone morphogenetic proteins (BMPs), which are involved in various cellular processes, including proliferation [17,18], differentiation [19,20], and apoptosis [21,22]. Through inhibition of BMP signaling, Noggin regulates tissue homeostasis and prevents pathological calcification [23,24,25]. This regulatory function suggests that Noggin is essential to maintaining the structural and functional integrity of the IVD. It is hypothesized that Noggin is a critical factor in maintaining IVD health, and decreased expression of Noggin in NPCs is linked to IVD degeneration, thus affecting the development of LBP.

This study investigated the effects of altered Noggin expression on healthy human NPCs. By exploring the relationship between Noggin levels and IVD health, we aim to elucidate the molecule’s role in disc degeneration and identify potential therapeutic targets for preventing or reversing degeneration.

## 2. Results

### 2.1. NPCs Express Elevated Levels of Noggin

Gene expression of human NPCs and donor-matched osteoblasts (OBs) or mesenchymal stem cells (MSCs) was conducted to characterize the expression of Noggin in various cell types. Specifically, the relative expression levels of Noggin were quantified. Expression of Noggin in NPCs was significantly higher compared to that of autologous human primary OBs (*p* < 0.05, Figure 1A). Similarly, NPCs trended toward a more elevated expression of Noggin than that of MSCs (Mean: NPC 2.35, MSC 0.87, *p* = 0.18; Figure 1B). Moreover, the average expression of Noggin in NPCs was 8.38 times higher than that in autologous OBs, and 1.47 times than that in MSCs. These results revealed that Noggin was highly expressed in human primary NPCs.

### 2.2. Noggin in Human Primary NPCs Was Effectively Knocked Down by Lentivirus (Lenti-shNoggin)

We silenced Noggin expression in human primary NPCs via lentiviral transduction (Lenti-shNoggin) by constructing a Noggin downregulated expression vector. To validate its effectiveness, HEK293 cells were infected with the target lentivirus. It showed that Noggin expression was significantly inhibited after transduction using Lenti-shNoggin (Figure 2A). Subsequently, lentiviral particles were used to transduce human primary NPCs. As in HEK293 cells, Noggin expression in NPCs was downregulated successfully (Figure 2B and Appendix A).

### 2.3. Silencing of Noggin in Human Primary NPCs Inhibited Cell Growth and Induced Apoptosis

To address the possible effects of Noggin knockdown on NPCs, an XTT assay was conducted to check the viability/proliferation of human NPCs transduced with Lenti-shNoggin. The results revealed that Noggin knockdown was significantly correlated with proliferation suppression in NPCs (Figure 3A). After 10 days of culture, the NPCs in the control group showed active growth compared to Day 0 (*p* < 0.05), whereas the NPCs in the Lenti-shNoggin group exhibited a slower proliferation rate (*p* = 1.00). To understand the cause of decreased NPCs proliferation after silencing Noggin, cells were stained with Hoechst 33258 dye. In the Lenti-shNoggin group, cells reflected induced chromatin condensation and nuclear shrinkage or fragmentation, and increased apoptosis (Figure 3B). Compared to the control group (apoptosis% = 1.34%), the percentage of cell apoptosis in the Lenti-shNoggin group reached 22.93%, significantly larger than that of the control group (*p* < 0.05, Figure 3C). These findings were further supported by the results of the colony formation assay, which indicated that silencing Noggin decreased colony formation ability in vitro (Figure 4A). There was a significantly higher number of colonies formed in the control group compared to the Lenti-shNoggin group (*p* < 0.01, Figure 4B). These results suggest that Noggin may have a critical role in maintaining the homeostasis of NPCs.

### 2.4. Inhibition of Noggin Expression Affected the Cell Viability of Human Primary NPCs

To further assess the effect of Noggin inhibition on the survival of NPCs, cell viability was analyzed. Results indicated a positive correlation between the time after transduction and the extent of cell death. On Day 0, cell viability was not significantly different between groups (*p* = 0.40). Generally, the cell death rate in the control group was stable over the culture period. However, the NPCs in the Lenti-shNoggin group presented an increase in cell death rate over time (Figure 5A,B. Day 0, 4.75%; Day 4, 7.12%; Day 8, 10.99% in Lenti-shNoggin group). Further, cell viability was significantly decreased after transduction in comparison to the control on Day 4 (*p* < 0.01) and Day 8 (*p* < 0.05).

### 2.5. Knockdown of Noggin in Human NPCs Could Hinder Cellular Anabolism and Enhance Catabolism

Anabolic and catabolic markers of NPCs were quantified at the transcript level by RT-qPCR to determine the role of Noggin in regulating the metabolism of human NPCs. Aggrecan core protein (*ACAN*) and type II collagen (*COL2*), anabolic markers of NPCs, were downregulated following Noggin knockdown (Mean: *ACAN* 0.50, *p* < 0.01; *COL2* 0.54, *p* = 0.20). In contrast, type I collagen (*COL1*), a fibrotic marker, was upregulated with Noggin knockdown (Mean = 1.38, *p* = 0.13). These results indicate that Noggin expression is essential for maintaining the NPCs phenotype (Figure 6A). Conversely, downregulated Noggin accelerated the process of catabolism in NPCs. The average expression of matrix metalloproteinase 3 (*MMP3*) and a disintegrin and metalloproteinase with thrombospondin motifs 4 (*ADAMTS4*) among the measured six donors increased (Mean: *MMP3* 25.54, *p* = 0.16; *ADAMTS4* 1.91, *p* = 0.48), especially *MMP3*. Unexpectedly, the expression of matrix metalloproteinase 13 (*MMP13*) was not consistent with the result of the other catabolic markers (Mean = 0.31, *p* < 0.0001) and was decreased in comparison to that of the control; the overall catabolic effect remained that the increase was more significant than the decrease (Figure 6B).

## 3. Discussion

The degeneration of the IVD is a pivotal factor in the development of LBP, and is a significant global health issue [26]. Understanding the molecular mechanisms underlying IVD degeneration is crucial for developing effective therapeutic strategies.

In this study, Noggin was highly expressed in healthy IVD NPCs compared to autologous OBs/MSCs. Previous studies [16] from our group involving smaller sample sizes of various IVD-derived cell types have demonstrated similar trends, further illustrating the unique role of Noggin in healthy IVD. However, whether Noggin is a critical factor in the transition from healthy to degenerated IVD remains unclear. To investigate this, the lentivirus-mediated knockdown of Noggin expression in human NPCs was used to explore the impact of Noggin on NPCs survival.

As a fundamental aspect of cellular biology, cell growth refers to the increase in cell size and number [27]. In the context of NPCs within the IVD, maintaining robust cell growth is indispensable for the health and functionality of the IVD [28]. The NP serves as the central gelatinous core of the IVD, providing mechanical support and flexibility to the spine [29]. NPCs are responsible for producing and maintaining the ECM [30], a complex network of proteins and polysaccharides essential for the structural integrity of the NP [31]. This study demonstrated that lentivirus-mediated downregulation of Noggin in human NPCs significantly suppressed cell growth. Noggin, a known antagonist of BMPs [32,33], plays an essential role in regulating BMP signaling pathways involved in cellular proliferation and differentiation [17], contributing to bone formation and regeneration [34]. By inhibiting BMPs, Noggin ensures the proper balance of cellular activities necessary for tissue homeostasis [35]. Interestingly, the suppression of cell growth observed in our study suggests that Noggin is vital for maintaining the proliferative capacity of NPCs, which is fundamental for the continuous renewal and repair of the NP.

This study also highlighted the role of Noggin in regulating apoptosis in NPCs. The lentivirus-mediated knockdown of Noggin induced significant apoptosis in NPCs, as evidenced by an apoptosis staining assay. This result was consistent with the slow cell growth in the Lenti-shNoggin group. Apoptosis, or programmed cell death, is a tightly regulated process that eliminates damaged or dysfunctional cells [36]. However, excessive apoptosis can lead to tissue degeneration and loss of cellular function [37,38]. Studies [39,40] have shown that BMP-2 can induce apoptosis in various cell types, including MSCs. By antagonizing BMP-2, Noggin may help maintain cell viability by preventing apoptosis. These results suggest that reduced Noggin expression removes this protective effect, rendering NPCs more susceptible to apoptotic signals and decreasing cell growth.

Generally, as apoptosis continues, cell death also increases [36]. In Live/Dead staining, we found that the death rate of transduced NPCs on Day 4 and Day 8 was significantly higher than that of the control group. In addition to the effect of increased apoptosis, the reduced viability of NPCs may also be related to the disrupted balance between anabolic and catabolic activity observed in our study. Relative gene expression analysis revealed significant alterations in the expression levels of genes associated with these metabolic processes. The downregulation of Noggin hindered anabolic processes while enhancing catabolic activity in NPCs. This disruption in cellular metabolism likely contributes to IVD degeneration, as the balance between anabolic and catabolic activity is essential for maintaining extracellular matrix integrity and overall disc health. The increased apoptosis further supports that reduced Noggin expression leads to detrimental cellular outcomes in NPCs.

The colony formation assay is a widely used method for assessing cells’ clonogenic potential and proliferative capacity [41]. This assay measures the ability of a single cell to grow into a colony, reflecting the cell’s ability to undergo multiple rounds of division and proliferation [42]. In this study, the downregulation of Noggin in NPCs significantly decreased colony formation, indicating a diminished proliferative capacity. Further, the reduced colony formation is associated with the increased apoptosis observed in this study. Apoptosis minimizes the number of viable cells available to form colonies, directly impacting the clonogenic potential of NPCs.

Moreover, the enhanced catabolic activity observed after Noggin downregulation may contribute to reduced colony formation by degrading ECM components necessary for cell attachment and growth [43]. Studies [44,45] have shown that ECM integrity is critical for supporting cell proliferation and colony formation. Thus, the disrupted balance between anabolic and catabolic activity in NPCs with reduced Noggin expression likely creates a less favorable microenvironment for colony formation.

Overall, Noggin plays a central role in keeping the health of NPCs within IVD by regulating BMP signaling and cellular homeostasis. Reduced expression of Noggin in NPCs leads to increased BMP-2 activity, which induces osteogenic differentiation, while simultaneously promoting apoptosis and inhibiting NPCs proliferation. To some extent, it disrupts the balance between anabolic (constructive) and catabolic (destructive) processes. Healthy NPCs maintain the production of ECM proteins like ACAN and COL2, which are vital for sustaining the phenotype of the IVD. While lower Noggin levels alter the NPCs’ microenvironment, increasing the expression of catabolic enzymes such as MMP3 and ADAMTS4. These enzymes degrade the ECM of NP and weaken the structural integrity, ultimately contributing to the degeneration of the disc.

Despite the promising findings, this study has several limitations that should be acknowledged. The use of in vitro models, while valuable for controlled mechanistic studies, may not fully replicate the complex environment of the human IVD in vivo. The effects of Noggin knockdown observed in cultured NPCs may differ from those in a living organism, where multiple cell types and systemic factors interact. Furthermore, the study focused on primary NPCs from a limited number of donors, which may not capture the full variability in Noggin expression and function across different individuals.

## 4. Materials and Methods

### 4.1. Human Materials and Cell Isolation

Human material, including bone fragments, intervertebral disc tissues, and human bone marrow aspirates, were obtained from patients undergoing spinal surgery at the Inselspital University Hospital Bern with written consent. Six donor-derived NPCs were used for transduction, while ten donors were used for the comparison of Noggin expression between NPCs and autologous MSCs/OBs (Table 1 and Table 2). All donor tissues and cells were collected anonymously and with written consent. Approvals were under the Insel University Hospital’s general consent, including anonymizing biological material and health-related data. The injured vertebral body bone fragments were cut into smaller pieces, 3–5 mm in diameter. The pieces were washed with phosphate-buffered saline (PBS) before being transferred to T75 flasks for culturing in alpha-minimum essential medium (α-MEM) containing 10% fetal bovine serum (FBS, Sigma-Aldrich, Inc, Buchs, Switzerland) and 1% penicillin/streptomycin (P/S, Sigma-Aldrich, Inc, Buchs, Switzerland). Primary OBs were expanded via the active outgrowth technique and selected for plastic adherence. The medium was replaced after roughly one week of culture, coinciding with the observation of multiple initial OB populations. IVD tissues were processed within 24 h following surgery, with the NP being harvested from the IVD tissue, usually in the operating room by a skilled spine surgeon. These tissues were then sequentially digested using 1.90 mg/mL pronase (Roche, Basel, Switzerland) for one hour, followed by collagenase II (64 U/mL; Worthington, London, UK) on a plate shaker at 37 °C overnight. To remove residual fragments, the digested tissue mixture was filtered through a 100 μm cell strainer (Falcon, Becton Dickinson, Allschwil, Switzerland). The obtained NPCs were cultured in low-glucose (1 g/L) Dulbecco’s Modified Eagle Medium (LG-DMEM, Gibco, Life Technologies, Zug, Switzerland) containing 10% FBS and 1% P/S. MSCs were isolated from bone marrow samples aspirated from vertebrae during spinal surgery (5–10 mL) using gradient centrifugation (Histopaque-1077, Sigma-Aldrich) [46]. These MSCs were subsequently expanded in α-MEM (from Sigma-Aldrich) with 10% FBS, 1% P/S, and 2.50 ng/mL basic fibroblast growth factor 2 (bFGF2, Peprotech, London, UK) [47].

### 4.2. Target Cells Transduction

To generate Noggin knockdown NPCs, lentiviral vector particles with shRNA targeting Noggin were used to transduce primary NPCs (Noggin shRNA plasmid, with GFP, OriGene Technologies, Inc., Rockville, MD, USA; and the shRNA sequences are listed in Table 3). Lentiviral vector particles were produced by co-transfection of the vector construct and packaging constructs pMDLg/pRRE, pRSV-Rev, and pCMV-VSV-G in 293T cells with lipofectamine LTX (Invitrogen, Carlsbad, CA, USA), as previously described [48]. Then the collected virus particles were used for transduction of primary NPCs.

### 4.3. Cell Proliferation Assay

NPCs from two groups were seeded into 96-well plates (2000 cells/well). They were assessed using a reagent kit for the quantification of cell proliferation (TACS^®^ XTT, R&D Systems, Inc., Minneapolis, MN, USA) according to the manufacturer’s instructions. At days 0, 4, and 8 of culture, cells were incubated with the XTT mixture for 3 h at 37 °C. Subsequently, absorbance was measured at 450 nm using a microplate reader (SpectraMax M5, Bucher Biotec, Basel, Switzerland).

### 4.4. Apoptosis Analysis

Hoechst 33258 (from Sigma-Aldrich) was utilized to detect the apoptosis in human NPCs after transduction [49,50]. Cells were seeded into 6-well plates (3000 cells/well) and cultured for five days. Subsequently, the medium of all wells was removed and 2 mL 1× PBS with a final concentration of 10 μg/mL Hoechst 33258 was added inside. Finally, they were incubated at 37 °C for 10 min before imaging.

### 4.5. Live/Dead Staining

To assess the cell viability, NPCs from each group were seeded into 24-well plates (8000 cells/well) and cell viability was checked at days 0, 4, and 8. Briefly, NPCs were immersed into serum-free medium containing 5 µM calcein-AM (#17783-1MG; Sigma-Aldrich) to stain the living cells and 1 µM ethidium homodimer (#46043-1MG-F; Sigma-Aldrich) to stain the dead cells and incubated at 37 °C. After 30 min of incubation, images were taken under a fluorescence microscope. Finally, the living and dead cells were quantified using a custom-made macro for ImageJ software version 1.52 [51].

### 4.6. Colony Formation Assay

Cell suspensions (2000 cells/well) were seeded into a 6-well plate and cultured in a cell culture incubator. After 2 weeks, the cell colonies were washed thrice with 1× PBS. The colonies were then fixed with 4% paraformaldehyde for 30 min and stained with 0.2% crystal violet (Millipore, Boston, MA, USA) for 30 min. Colonies in each group were subsequently counted.

### 4.7. RNA Extraction and Relative Gene Expression by RT-qPCR

Total RNA was extracted from human NPCs of both groups, evaluating anabolic and catabolic marker gene expression after transduction, including *ACAN*, *COL1*, *COL2*, *MMP3*, *MMP13*, and *ADAMTS4*. Relative gene expression was normalized to the reference gene GAPDH, and then relative to control. Cells from the same donor were harvested for RNA extraction to evaluate Noggin expression between human OBs/MSCs and NPCs. The expression levels in NPCs were subsequently compared to those in OBs/MSCs, serving as the control. RNA isolated from the samples was converted into cDNA using the High-Capacity cDNA Reverse Transcription kit (#4368814; Thermo Fisher Scientific). The cDNA was mixed with iTaq Universal SYBR Green Supermix (#1725122; Bio-Rad, inc.) and human-specific oligonucleotide primers listed in Table 4. RT-qPCR analysis was performed using the CFX96™ Real-Time System (#185-5096; Bio-Rad, inc.). Relative gene expression was analyzed via the 2^−ΔΔCt^ method [52], with *GAPDH* serving as the reference gene.

### 4.8. Statistics

The statistical evaluation was performed by Student’s *t*-test between two groups and one-way ANOVA for three or more groups. The analyses were conducted using GraphPad Prism 10 (GraphPad Software, San Diego, CA, USA) software and a *p* value < 0.05 was considered statistically significant. All experiments were performed in triplicates. n = number of biological replicates is indicated in all graphs.

## 5. Conclusions

Our study demonstrates that low expression of Noggin in human NPCs inhibits cell growth and induces apoptosis. Suppression of Noggin results in the downregulation of anabolic processes and upregulation of catabolic processes in NPCs. These findings suggest that the decreased expression of Noggin is potentially linked to degeneration in human IVD. This research provides valuable insights into the role of Noggin in IVD homeostasis, highlighting its significance in maintaining disc health. Understanding Noggin’s function could inform the development of targeted therapies for preventing or treating LBP, thus contributing to improved patient outcomes and quality of life.

## Figures and Tables

**Figure 1 ijms-25-11803-f001:**
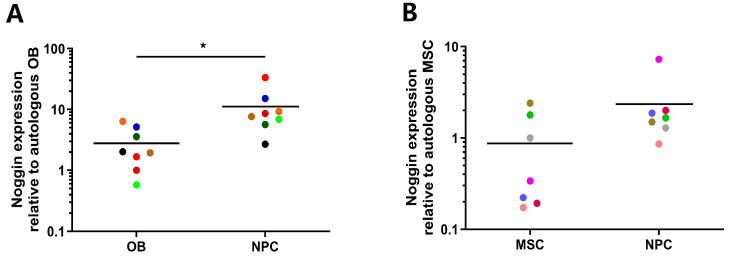
Elevated expression of Noggin in human NPCs relative to autologous OBs/MSCs: (**A**) Noggin was expressed more in human NPCs than in autologous OBs. (**B**) A similar increased trend of Noggin in NPCs compared to MSCs. (Identical colors indicate the same donor. Shown as means. *p*-value, * <0.05; *n* = 7–8).

**Figure 2 ijms-25-11803-f002:**
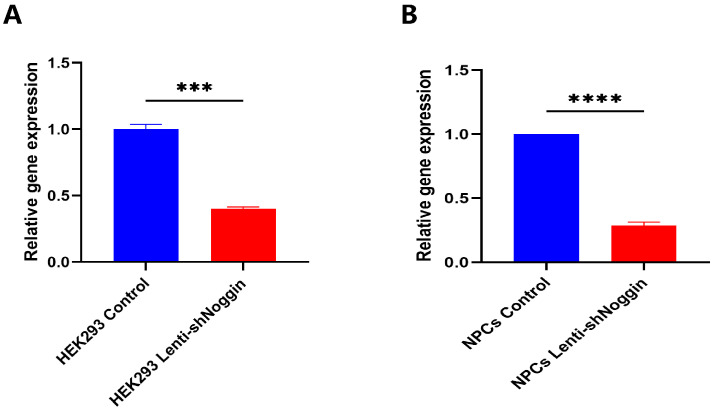
Transduction of HEK293 and NPCs with Lenti-shNoggin successfully: (**A**) RT-qPCR analysis of Noggin expression in HEK293 cells between the control and Lenti-shNoggin groups. (**B**) Noggin expression in human NPCs was suppressed after transduction. (Mean ± SEM. *p*-value, *** <0.001; **** <0.0001; *n* = 3).

**Figure 3 ijms-25-11803-f003:**
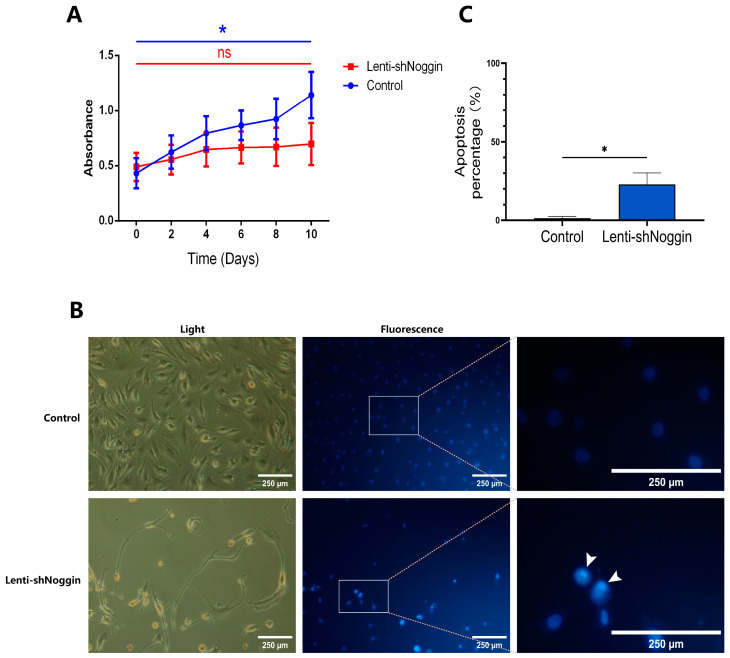
Low expression of Noggin in human NPCs induced apoptosis and inhibited cell growth: (**A**) The proliferation of human NPCs was limited when transduced by Lenti-shNoggin. (**B**) Quantification of cell apoptosis by Hoechst 33258 dye on microscopy images. (**C**) Quantitative analysis for apoptosis by nucleus counting. (Apoptosis under Hoechst 33258 dye: bright blue, as the white arrows indicate. Scale bar: 250 μm. Shown are Means ± SEM. *p*-value, ns >0.05; * <0.05; *n* = 3–5).

**Figure 4 ijms-25-11803-f004:**
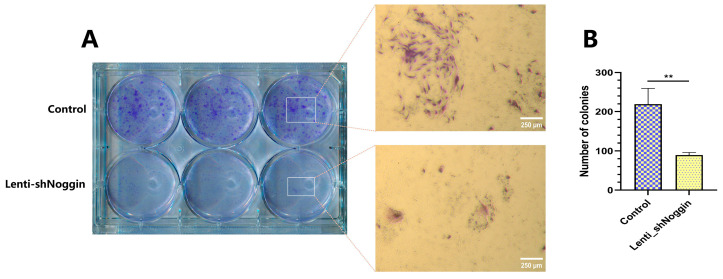
Colony formation assay revealed inhibition of human NPCs’ proliferative potential after transduction by Lenti-shNoggin: (**A**) Fewer cell clones were found in transduced human NPCs after culture for 14 days. (**B**) The number of cell clones was counted and reduced significantly in the Lenti-shNoggin group under the microscope. (Scale bar: 250 μm. Mean ± SEM. *p*-value, ** <0.01; *n* = 3).

**Figure 5 ijms-25-11803-f005:**
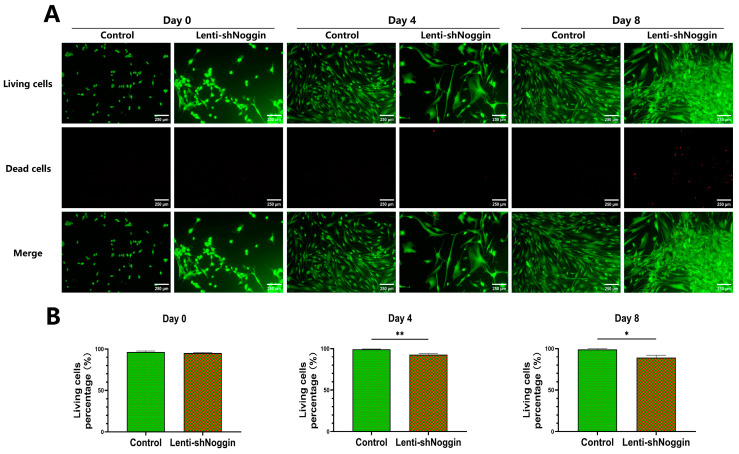
Cell viability analysis of human primary NPCs transduced with Lenti-shNoggin indicated an increase in cell death: (**A**) Calcein AM and Ethidium Homodimer-1 dye were used to stain the human NPCs after transduction. (**B**) The semi-quantitative analysis on cell viability about living cells for control and Lenti-shNoggin group. (Scale bar: 250 μm. Mean ± SEM. *p*-value, * <0.05; ** <0.01; *n* = 3).

**Figure 6 ijms-25-11803-f006:**
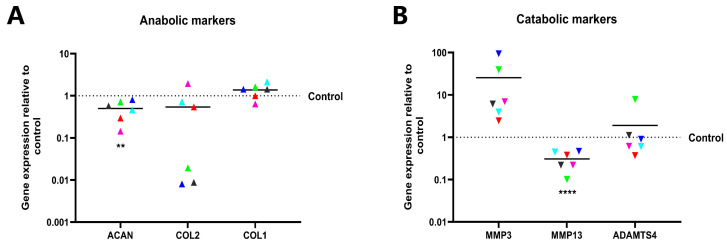
Single data diagrams illustrating the relative gene expression of Noggin in human NPCs could hinder cellular anabolism and enhance catabolism: (**A**) Suppression of Noggin resulted in a downregulation of the anabolic markers in human NPCs. (**B**) Catabolism in human NPCs after transduction. (Identical colors indicate the same donor. Shown as means. *p*-value, ** <0.01; **** <0.0001; *n* = 6).

**Table 1 ijms-25-11803-t001:** The NP tissues that were collected from human donors for NPC isolation and transduction in this study.

Donor	Age	Sex	Type	Level	NP Tissues	NPCs Isolation
1	36	M	T	L3-L4	√	√
2	30	M	T	L1-L2	√	√
3	32	M	T	L2-L3	√	√
4	30	M	T	T12-L1	√	√
5	50	M	T	T12-L1	√	√
6	20	M	T	L2-L3	√	√

Donor’s list of NP tissues, the NPCs in this study were isolated from these NP tissues. M, male; T, trauma; √, collected. Passage, 3–5.

**Table 2 ijms-25-11803-t002:** The details of donors for comparison of Noggin expression between NPCs and autologous MSCs/OBs.

Donor	Age	Sex	Type	NP Tissues	Level	Bone Marrow	Bone Fragments
1	36	F	T	√	T12-L1	x	√
2	36	M	T	√	L2-L3	√	√
3	74	F	T	√	L1-L2	x	√
4	39	F	T	√	T12-L1	x	√
5	30	M	T	√	L1-L2	√	√
6	77	F	T	√	L1-L2	√	√
7	32	M	T	√	L2-L3	√	√
8	30	M	T	√	T12-L1	√	√
9	78	M	D	√	L3-L4	√	x
10	63	M	D	√	L3-L4	√	x

F, female; M, male; T, trauma; D, degeneration; √, collected; x, not collected. Passage, 2–4.

**Table 3 ijms-25-11803-t003:** The shRNA sequences to target Noggin.

shRNA	Primer Sequence
Noggin	5′ GCT GCG GAG GAA GTT ACA GAT GTG GCT GT 3′

**Table 4 ijms-25-11803-t004:** Human gene primers for RT-qPCR.

Gene	Accession No.	Forward Sequence	Reverse Sequence
*GAPDH*	NM_001289745.2	ATC TTC CAG GAG CGA GAT	GGA GGC ATT GCT GAT GAT
*NOG*	NM_001078309.1	CAG CAC TAT CTC CAC ATC CG	CAG CAG CGT CTC GTT CAG
*ACAN*	NM_001135.4	CAT CAC TGC AGC TGT CAC	AGC AGC ACT ACC TCC TTC
*COL1*	NM_000089.3	GTG GCA GTG ATG GAA GTG	CAC CAG TAA GGC CGT TTG
*COL2*	XM_017018831.3	AGC AGC AAG AGC AAG GAG AA	GTA GGA AGG TCA TCT GGA
*MMP3*	NM_002422.5	CAA GGC ATA GAG ACA ACA TAG A	GCA CAG CAA CAG TAG GAT
*MMP* *13*	NM_002427.4	AGT GGT GGT GAT GAA GAT	CTA AGG TGT TAT CGT CAA GTT
*ADAMTS4*	XM_054339708.1	TTC CTG GAC AAT GGC TAT GG	GTG GAC AAT GGC GTG AGT

*GAPDH*, glyceraldehyde-3-phosphate dehydrogenase; *NOG*, noggin; *ACAN*, aggrecan core protein; *COL1*, type I collagen; *COL2*, type II collagen; *MMP3*, matrix metalloproteinase 3; *MMP13*, matrix metalloproteinase 13; *ADAMTS4*, a disintegrin and metalloproteinase with thrombospondin motifs 4. Primer sequence: 5′–3′.

## Data Availability

All the data used in this research are available on request from the first and corresponding author.

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
