# Peer review of "The Role of the Bone Morphogenetic Protein Antagonist Noggin in Nucleus Pulposus Intervertebral Disc Cells"

_ijms, 2024, doi:10.3390/ijms252111803_

Round 1
Reviewer 1 Report
Comments and Suggestions for Authors
I think it is a very interesting artcile. Please check my comments below.
Results paragraph
1. Please be consistent as the decimal places are probably not consistent.
2. P-values are not specifically indicated in some places. If you recognize a significant difference, please be sure to state it.
3. Figure 6: Are the magnification magnitudes of the control and Lenti-shNoggin groups equal?Why does the Lenti-shNoggin group appear larger? Please confirm.
4. Figure 7: You state that ACAN and COL2, markers of NP cells, were downregulated following Noggin knockdown.In contrast, COL1, a fibroblastic marker, was upregulated with In contrast, COL1, a fibroblastic marker, was upregulated with Noggin knockdown.
Can you please provide specific mean and P values? Could you also show * in the graphs for COL2 and COL1?
5. Figure 7: You state that downregulated Noggin accelerated the process of catabolism in NPCs. But MMP13 is significantly decreased. You need to mention whether there is any inconsistency in your results in this regard. Also, the P value of each gene needs to be stated.
6. Figure 7: Check to see if you need to compare the expression of genes associated with BMPs.
Materials and Methods paragraph
1. Please provide the specific age of the donor, not the Year of birth.
2. Is the notation “collagen 2” scientifically appropriate?” Type II Collagen “ might be better. Please confirm.
3.Check to see if there was any difference in the degree of degeneration of the disc to be retrieved or if it is not necessary to mention it.
Because different degrees of degeneration have different effects on BMPs.
Author Response
Reply to Reviewer 1
Question: Q Answer: A
Results paragraph
Q1. Please be consistent as the decimal places are probably not consistent.
A1. Thank you for your suggestion. Currently all data in the manuscript are kept to two decimal places.
Q2. P-values are not specifically indicated in some places. If you recognize a significant difference, please be sure to state it.
A2. Thank you for your suggestion. P-values ​​are now indicated in the corresponding positions in the results section of the manuscript.
Q3. Figure 6: Are the magnification magnitudes of the control and Lenti-shNoggin groups equal? Why does the Lenti-shNoggin group appear larger? Please confirm.
A3. This is a very meaningful question. They seem to have different magnifications, but in fact, they have the same magnification, both using a 5x objective lens. When cells have enough space to grow, they will fully expand and look bigger. When the growth space is small and there are many cells, the cells often appear small due to space limitations.
Q4. Figure 7: You state that ACAN and COL2, markers of NP cells, were downregulated following Noggin knockdown. In contrast, COL1, a fibroblastic marker, was upregulated with In contrast, COL1, a fibroblastic marker, was upregulated with Noggin knockdown.
Can you please provide specific mean and P values? Could you also show * in the graphs for COL2 and COL1?
A4. Thank you for your question. The relevant mean values ​​and P values ​​now are added to the corresponding positions in this section.
For your second question, among all the tested donors, although the COL2 expression level of most donors was lower than that of the control group (Mean = 0.54, p = 0.20), and the COL1 expression level was higher than that of the control group (Mean = 1.38, p = 0.13), the corresponding P values were still greater than 0.05 due to the large variance. Therefore, the * was not marked in the figure.
Q5. Figure 7: You state that downregulated Noggin accelerated the process of catabolism in NPCs. But MMP13 is significantly decreased. You need to mention whether there is any inconsistency in your results in this regard. Also, the P value of each gene needs to be stated.
A5. Thank you for your good question. In terms of the overall effect, reducing the expression of Noggin in NPCs accelerated its catabolism. Enhanced catabolism does not mean that the expression of every marker should increase. The decrease in MMP13 may be that the increase in MMP3 triggers the negative feedback regulation of MMP13 to avoid excessive matrix degradation. It may also be that COL2 synthesis is reduced, and negative feedback leads to a corresponding decrease in the synthesis of its main decomposer MMP13.
For the second question, according to your suggestion, we have now given the corresponding P value in this section.
Q6. Figure 7: Check to see if you need to compare the expression of genes associated with BMPs.
A6. Because the topic of this manuscript is about the effect of Noggin on cell survival, the content of the manuscript mainly revolves around this topic, so we did not test BMPs related genes.
However, this is a very good suggestion, and we will seriously consider this suggestion in future in-depth research.
Materials and Methods paragraph
Q7. Please provide the specific age of the donor, not the Year of birth.
A7. We have followed your suggestion and now changed the year of birth to age. Thank you.
Q8. Is the notation “collagen 2” scientifically appropriate?” Type II Collagen “ might be better. Please confirm.
A8. We think this is a good suggestion and have changed collagen 2 to Type II collagen in the latest manuscript.
Q9. Check to see if there was any difference in the degree of degeneration of the disc to be retrieved or if it is not necessary to mention it.
Because different degrees of degeneration have different effects on BMPs.
A9. Because the donors used for the main experiments in this study were all from trauma, and the only two degenerative donors in Figure 1/Table 2 were used to detect the expression difference of Noggin between NPCs and OBs/MSCs. No research on BMPs was involved. Therefore, we think it is unnecessary to mention it.
Reviewer 2 Report
Comments and Suggestions for Authors
In this study the authors attempt to assess the role of Noggin silencing in intervertebral disc degeneration. I am afraid the study presents only limited and preliminary information that remains at a descriptive level, with no functional experiments conducted and no mechanistic aspect.
1. The manuscript contains many grammar and syntax errors that need to me corrected (for examples, please refer to the uploaded file).
2. Ln28: I feel it is preferable to avoid the term CFU, which is usually associated with microbiology. Colony formation assay would be more appropriate here.
3. Throughout the text: The authors should replace PCR or RT-PCR by RT-qPCR.
4. Abbreviations OBs and MSCs should be defined the first time they appear in the text.
5. I feel that Figures 1C and D are redundant to Figures 1A and B and should be omitted.
6. Figures 2A and 3A present entirely technical data and in my opinion should move to the supplemental material.
7. The way they present their data, the authors give the impression to have confused the terms viability, proliferation, apoptosis, cell death. Data presented in Figure 4C is incompatible with data presented in Figure 4A, in which no cell death is shown. If Noggin silencing was toxic, then absorbance at days 2-10 (representing cell number) would be lower than that at day 0. Furthermore, in Figure 6, why did not the authors assess cell death also at day 10? In Figure 4C, the authors refer to an apoptosis index approx. 20%. This percentage cannot exceed the percentage of cell death (that includes apoptosis and necrosis). If cell death is exclusively apoptotic, then these two percentages should be at least equal.
8. Observing cellular morphology presented in Figures 4 and 5, cells seem to have an enlarged and irregular shape. Is there any chance Noggin knocking-down leads cells to senescence?

The manuscript contains many grammar and syntax errors that need to me corrected (for examples, please refer to the uploaded file).
Author Response
Reply to Reviewer 2
Question: Q Answer: A
In this study the authors attempt to assess the role of Noggin silencing in intervertebral disc degeneration. I am afraid the study presents only limited and preliminary information that remains at a descriptive level, with no functional experiments conducted and no mechanistic aspect.
Q1. The manuscript contains many grammar and syntax errors that need to me corrected (for examples, please refer to the uploaded file.
A1. Thank you for your suggestion. Currently we revise it carefully and upload it again
Q2. Ln28: I feel it is preferable to avoid the term CFU, which is usually associated with microbiology. Colony formation assay would be more appropriate here.
A2. Thank you for your suggestion. We have now changed CFU to colony formation assay.
Q3. Throughout the text: The authors should replace PCR or RT-PCR by RT-qPCR.
A3. We have now modified this in the latest version of the manuscript based on your suggestion.
Q4. Abbreviations OBs and MSCs should be defined the first time they appear in the text.
A4. Thanks for your reminder. We forgot that the “Materials and Methods” section was placed at the end instead of second. We have now changed this in the latest version.
Q5. I feel that Figures 1C and D are redundant to Figures 1A and B and should be omitted.
A5. Thank you for your question. Figures 1C and D have now been deleted, leaving only Figures 1A and B.
Q6. Figures 2A and 3A present entirely technical data and in my opinion should move to the supplemental material.
A6. Thank you. We have now reintegrated the original Figures 2 and 3 into a new Figure 2, and moved Figures 2A and 3A to the supplemental material.
Q7. The way they present their data, the authors give the impression to have confused the terms viability, proliferation, apoptosis, cell death. Data presented in Figure 4C is incompatible with data presented in Figure 4A, in which no cell death is shown. If Noggin silencing was toxic, then absorbance at days 2-10 (representing cell number) would be lower than that at day 0. Furthermore, in Figure 6, why did not the authors assess cell death also at day 10? In Figure 4C, the authors refer to an apoptosis index approx. 20%. This percentage cannot exceed the percentage of cell death (that includes apoptosis and necrosis). If cell death is exclusively apoptotic, then these two percentages should be at least equal.
A7. This is a very good question. Thank you.
- Because the NPCs we obtained after transduction are not pure, there are some NPCs that have not been successfully transfected. These unsuccessfully transduced NPCs continued to proliferate, so the absorbance on day 2-10 is higher than that on day 0.
- The reason why Figure 6 (now the latest version has been adjusted to Figure 5) did not detect cell death on day 10 is that because when we considered the time for changing the cell medium, compared to setting the detection time points on day 0, 4, and 8, the setting of day 0, 5, and 10 would make the time interval between the two time points for changing the medium too long, and we were worried that the cell growth would be affected, or even died, so we finally chose to set day 0, 4, and 8 as the detection time points.
- During apoptosis, cells undergo morphological changes, such as cell shrinkage, nuclear fragmentation, and apoptotic body formation, but cells may still maintain a certain degree of membrane integrity during this stage, especially in the early stages of apoptosis, where these cells have not yet entered the final state of disintegration or necrosis, and the dye (Ethidium Homodimer-1 dye) cannot pass through the intact cell membrane, so they are not marked as dead cells in death detection, resulting in a higher apoptosis rate and a lower overall mortality rate.
Q8. Observing cellular morphology presented in Figures 4 and 5, cells seem to have an enlarged and irregular shape. Is there any chance Noggin knocking-down leads cells to senescence?
A8. When we inhibited the expression of Noggin in NPCs, we observed that NPCs grew slowly or even died. We are not sure whether this result is caused by cell senescence or not. But this is undoubtedly a very good question and reminder, because in future in-depth research, we can continue to explore this aspect.